# Validation of the Fast-Track Model: A Simple Tool to Assess the Severity of Diabetic Foot Ulcers

**DOI:** 10.3390/jcm12030761

**Published:** 2023-01-18

**Authors:** Marco Meloni, Benjamin Bouillet, Raju Ahluwalia, Juan Pedro Sanchez-Rios, Elisabetta Iacopi, Valentina Izzo, Chris Manu, Vouillarmet Julien, Claas Luedmann, José Luis Garcia-Klepzig, Jérome Guillaumat, Jose Luis Lazaro-Martinez

**Affiliations:** 1Diabetic Foot Unit, Department of System Medicine, University of Rome Tor Vergata, 00133 Rome, Italy; 2Endocrinology Department, University Hospital Center, 21000 Dijon, France; 3INSERM-University of Bourgogne Franche-Comté, LNC UMR1231, 21000 Dijon, France; 4Department of Trauma and Ortophaedic, King’s College Hospital, London SE5 9RS, UK; 5Diabetic Foot Unit, Vascular Surgery Department, Hospital Fundación Alcorcon, 28922 Madrid, Spain; 6Diabetic Foot Section, University of Pisa, Ospedale di Cisanello, 56124 Pisa, Italy; 7UniCamillus-Saint Camillus International, University of Health Science, 00131 Rome, Italy; 8Diabetic Foot Clinic, King’s College Hospital, Denmark Hill, London SE5 9RS, UK; 9Endocrinology Department, University Hospital Center, 69007 Lyon, France; 10Franziskus Krankenhaus Berlin, 10787 Berlin, Germany; 11Internal Medicine Department, Hospital Clinico San Carlos De Madrid, 28040 Madrid, Spain; 12Department of Vascular Surgery, University Hospital of Caen, 14033 Caen, France; 13Diabetic Foot Unit, Universidad Complutense de Madrid, 28040 Madrid, Spain

**Keywords:** diabetes, diabetic foot, fast-track, limb salvage, wound healing

## Abstract

This study aimed to validate the association between the grades of severity of diabetic foot ulcers (DFUs) identified by the fast-tack model and specific outcomes. Three hundred and sixty-seven patients with new DFUs who were referred to a tertiary level diabetic foot service serving Rome, Italy, were included. The fast-track model identifies three levels of DFUs’ severity: uncomplicated DFUs, including superficial wounds, not-infected wounds, and not-ischemic wounds; complicated DFUs, including ischemic wounds, infected wounds, and deep ulcers involving the muscles, tendons, or bones, and any kind of ulcers in patients on dialysis and/or with heart failure; and severely complicated DFUs, including abscesses, wet gangrene, necrotizing fasciitis, fever, or clinical signs of sepsis. Healing, minor and major amputation, hospitalization, and survival after 24 weeks of follow-up were considered. Among the included patients, 35 (9.6%) had uncomplicated DFUs, 210 (57.2%) had complicated DFUs, and 122 (33.2%) had severely complicated DFUs. The outcomes for patients with uncomplicated, complicated, and severely complicated DFUs were as follows, respectively: healing, 97.1%, 86.2%, and 69.8%; minor amputation, 2.9%, 20%, and 66.4%; major amputation, 0%, 2.9%, and 16.4%; hospitalization, 14.3%, 55.7%, and 89.3%; survival, 100%, 96.7%, and 89.3%. DFU severity was an independent predictor of healing, amputation, hospitalization, and survival. The current study shows an association between the grade of severity of DFUs identified by the fast-track model and the considered outcomes. The fast-track model may be a useful tool for assessing the severity and prognosis of DFUs.

## 1. Introduction

The delayed referral of persons with diabetic foot ulcerations (DFUs) to specialized diabetic foot services (DFSs) is a common issue worldwide [1,2]. In a study conducted in four European countries, in approximately 60% of cases, the duration of the DFUs was unknown or the diagnosis was delayed more than three weeks from the onset of the wounds [3]. In addition, health care providers working in primary care, e.g., general practitioners (GPs), reported that, in 40% of cases, they were not adequately trained in the management of DFUs; 30% of GPs were not able to assess the wounds or to grade their severity [4]. Currently, several DFU classifications are used differently, according to local policies or individual experiences [5]. Until a few years ago, no specific DFU description addressed the needs of health care professionals (HCPs) working in primary care to carry out easy wound assessments or define wound severity.

Therefore, the International Diabetic Foot Care Group (IDFCG), in cooperation with D-Foot International, developed the fast-track pathway (FTP) [6], a clinical tool that helps in identifying wound severity based on clinical features, with the aim of reducing late referrals to specialized diabetic foot services (DFSs). The FTP model identifies three levels of DFU severity. For each grade of severity, a specific timing of referral to a specialized DFS has been established [6]. The FTP has been accepted by the scientific community as a practical model to identify a specific pathway according to a DFU’s severity (see https://d-foot.org/projects/fast-track-pathway-for-diabetic-foot-ulceration, accessed on 24 October 2019), and its effectiveness has been recently documented [7].

However, the association between the grade-of-wound severity identified by the fast-track model and the DFU outcomes is not well-documented. In the current manuscript, the authors aimed to evaluate and validate the association between the grade-of-wound severity identified by the fast-track model and specific outcomes, including ulcer healing, minor and major amputation, hospitalization, and survival.

## 2. Materials and Methods

### 2.1. Patient Selection 

The current retrospective study follows patients with newly active DFUs who were referred to a tertiary level DFS serving Rome, Italy, from January 2019 to March 2022. Patients presenting with other conditions, stable treated ulcers, and/or reduced life expectancy (<6 months) related to their general health status were excluded.

All patients were managed by a limb-salvage protocol, following International Working Group on the Diabetic Foot (IWGDF) guidelines [8], including restoration of foot perfusion for peripheral ischemia, antibiotic therapy (and surgery if required) for infection, offloading of affected limbs, ulcer debridement, and local wound care based on best-evidence recommendations and the management of diabetes and associated comorbidities. At the time of admission, demographic and clinical data, as well as DFU characteristics, were recorded.

### 2.2. Clinical Assessment

#### 2.2.1. Comorbidities

Hypertension was considered in the case of current antihypertensive therapy. Hypercholesterolemia was considered in cases of current statin therapy. Ischemic heart disease (IHD) was considered in cases of previous acute coronary syndrome or coronary revascularization, evidence of angina, significant changes on electrocardiography (above or under-levelling ST, q wave, inversion of T wave, or a new left bundle branch block). Heart failure (HF) was considered in the cases of recognized typical symptoms and signs of HF and a reduced left ventricular ejection fraction (LVEF) (<40%), normal or mildly reduced LVEF and elevated levels of brain natriuretic peptides (BNP > 35 pg/mL and/or NT-proBNP > 125 pg/mL), without a dilated left ventricle (LV), associated with relevant structural heart disease (LV hypertrophy/left atrial enlargement) and/or diastolic dysfunction [9]. Dialysis was considered in the case of end-stage renal disease (ESRD) requiring renal replacement therapy. Smoking was considered only for active smokers.

#### 2.2.2. Ulcer Characteristics and Treatment

Ulcer characteristics were reported at the time of presentation and first assessment at the DFS. Deep ulcers were considered as full thickness skin lesions, extending from the subcutaneous region to tendons, muscles, or bones. A diagnosis of infection was defined according to IWGDF guidelines [8]. Standard treatment of infection followed IWGDF recommendations, with initial broad spectrum antibiotic therapy and then further treatment according to culture results if required [8].

All patients received offloading for relieving pressure and trauma in the ulcer area, according to ulcer location, the presence of ischemia or infection (isolated or in conjunction), and individual needs [8]. Lower limb ischemia was defined as either no palpable distal pedal pulses, TcPO2 < 30 mmhg [8], and/or arterial stenosis/occlusions documented by duplex ultrasound (or computed tomography or MRI, if needed), requiring lower-limb revascularization. The revascularization procedure was performed for foot conditions, affected vessels, and a patient’s general condition, either by an endovascular or surgical (bypass) procedure [8,10].

### 2.3. Assessment of DFU Severity: The Proposed Model of DFU Description

The fast-track model provides specific clinical items for assessing the severity of each DFU [6]: patient factors, defined by the presence of concomitant comorbidities, such as ESRD and heart failure; general status of health, including the presence of fever or clinical signs of sepsis; vascular status, indicated by the presence of foot pulses; and ulcer factors, including depth, infection, presence of necrosis, presence of gangrene, or presence of an abscess.

Accordingly, three different levels of DFU severity were identified (see Table 1):

Uncomplicated DFUs were considered in cases of superficial ulcers, not-infected ulcers, or not-ischemic ulcers;

Complicated DFUs were considered in cases of ischemic (or suspected ischemic), infections (mild/moderate), deep wounds (involving soft tissue and/or bone), or any kind of ulcer in patients on dialysis or with heart failure;

Severely complicated DFUs were considered in cases of an abscess, the presence of wet gangrene, the presence of necrotizing fasciitis, fever, or clinical signs of sepsis.

Importantly, the FTP process recommends [6] that patients with uncomplicated DFUs can be managed in the community and should be referred to specialized DFSs in cases of ulcer size reduction of less than 30% after 2 weeks of standard care, while patients with complicated DFUs should be referred within 4 days and patients with severely complicated DFUs should be referred within 24 h. In cases of suspected necrotizing fasciitis or clinical signs of sepsis, patients should be referred immediately for emergency treatment.

### 2.4. Clinical Outcomes

Completed ulcer healing, minor and major amputations, hospitalization, and survival after 6 months of follow-up were evaluated. Definitive ulcer healing was taken to be the complete epithelialization of the target wound and maintenance of the closed healed epithelized surface for a minimum of 2 weeks. Healing was defined as complete wound healing without a major amputation, including healing of the wounds of patients who required a minor amputation to achieve wound healing and patients who recovered without a minor amputation. Any amputations below the ankle (e.g., digital amputations, ray amputations, metatarsal amputations, Lisfranc amputations, or Chopart amputations) were considered a minor amputations. Any amputations above the ankle were considered a major amputations.

### 2.5. Statistical Analysis

Statistical analysis was performed by SAS (JMP12; SAS Institute, Cary, NC, USA). Continuous variables were expressed as the mean ± SEM. Comparisons between groups characteristics were made with an X^2^ test (frequency data) or ANOVA (continuous data). Univariate logistic regression analysis was performed for all potential predictors of our outcomes of interest (healing, major amputations, hospitalizations, and survival) and represented as univariate hazard ratios (HRs), along with the respective 95% CI. Following this, all potential predictors were entered simultaneously to form a multivariate logistic regression analysis. Hazard ratios and corresponding 95% confidence intervals were obtained from stratified Cox proportional-hazards models. *p* < 0.05 was considered statistically significant.

The variables used in the univariate analysis (and in the multivariate analysis) included all potential factors of outcomes already documented in previous literature and based on data recorded in the current study (age, peripheral arterial disease, infection, wound duration, wound size, wound depth, ischemic heart disease, heart failure, and end-stage renal disease).

## 3. Results

Three hundred and sixty-seven patients were included in the study: 35 (9.6%) had an uncomplicated DFU; 210 (57.2%) had a complicated DFU; 122 (33.2%) had a severely complicated DFU.

### 3.1. Demographics and Patient Ulcer Characteristics Related

The mean age of the patients was 69 ± 13 years. There was a prevalence of males. Most of the patients were affected by type 2 diabetes with a long diabetes duration (see Table 2). The majority of the patients reported an ischemic DFU, an infected DFU, or a deep (to the bone) DFU. Patients with severely complicated DFUs had longer diabetes duration and poorer metabolic control than patients with uncomplicated DFUs or complicated DFUs. In addition, patients with severely complicated DFUs reported more cases of infection, deep ulcers, gangrene, larger ulcers, and longer wound duration than patients with complicated DFUs, and higher rates of IHD and ESRD (see Table 2).

### 3.2. Outcomes

Overall, 81.7% of the patients healed without a major amputation, 33.8% had a minor amputation, 7.1% had a major amputation, 62.9% needed hospitalization, and 94.5% survived after 6 months of follow-up. (See Table 3).

Patients with severely complicated DFUs, in comparison with patients with complicated and uncomplicated DFUs, had a reduced chance of healing (69.8% vs. 86.2% and 97.1%, respectively, *p* < 0.0001), a higher risk of minor amputation (66.4% vs. 20% and 2.9%, respectively, *p* < 0.0001), a higher risk of major amputation (16.4% vs. 2.9% and 0%, respectively, *p* < 0.0001), higher rates of hospitalization (89.3% vs. 55.7% and 14.3%, respectively, *p* < 0.0001), and less chance of survival after 6 months of follow-up (89.3% vs. 96.7% and 100%, respectively, *p* < 0.0001). See Table 3.

Following multivariate analysis, DFU severity (severely complicated DFU), PAD, infection, and osteomyelitis were independent predictors of healing; DFU severity and ESRD were independent predictors of major amputation and survival; and DFU severity, heart failure, infection, PAD, sepsis, and wound duration (>4 weeks) were independent predictors of hospitalization. See Table 4 and Table 5.

## 4. Discussion

The current study shows that outcomes of patients with DFUs worsened with increases in wound severity, as defined by the fast-track model, which was expected. The three patterns of DFU severity defined by the fast-track model are associated with different prognostic markers, e.g., healing, amputation, hospitalization, and survival.

Severely complicated DFUs resulted in reduced rates of healing and were more likely to lead to minor amputations, major amputations, hospitalizations, and mortality than uncomplicated and complicated DFUs. A severely complicated DFU is an independent predictor of non-healing, major amputation, mortality, and hospitalization. Patients with severely complicated DFUs also have more significant comorbidities; they reported heart failure in approximately 25% of cases and ESRD in approximately 20% of cases, which were the highest among any grade of DFU severity. Patients affected by heart failure and ESRD requiring dialysis are usually fragile subjects and are characterized by a higher risk of amputation and mortality in comparison with patients with preserved heart and renal function, as reported in the literature [11,12]. Heart failure in patients with DFUs appears to be an independent predictor of middle- and long-term mortality [11,13,14], while dialysis is an independent risk factor for major amputation and below-the-ankle arterial disease, which is the most aggressive pattern of PAD [12,13,14,15,16].

For these reasons, patients with heart failure and patients requiring dialysis have been included as complicated DFUs in the fast-track model, regardless of the characteristics of the DFU [6]. Therefore, it is imperative that these specific comorbidities receive early clinical and vascular assessments. Heart failure has been shown in this study to be associated with hospitalization, while ESRD was independently associated with major amputation and mortality.

We observed that patients with severely complicated DFUs reported a higher rate of infection than did patients with complicated DFUs (100% vs. 78.1%), as well as a higher rate of deep ulcers involving the bone (95.9% vs. 62.4%). However, ischemia was relatively similar in these two groups (65.6% and 61.9%, respectively). As is well known, infection, ischemia, and wound depth (specifically in cases involving bones and osteomyelitis) are also wound-related factors that independently increase the risk of non-healing, minor amputation, major amputation, and hospitalization [17,18,19,20].

The current study showed that, in addition to an association with the grade of DFU severity, ischemia was independently associated with non-healing and hospitalization, while infection and osteomyelitis were independently associated only with non-healing. Unsurprisingly, the highest incidence of these specific wound characteristics was among patients with severely complicated DFUs, reflecting the worse outcomes in these subjects when compared with patients with complicated and uncomplicated DFUs.

Nonetheless, a specific note should be made about wound duration at the time of assessment. Patients with severely complicated DFUs reported longer times of wound duration in comparison with patients with complicated and uncomplicated DFUs, and it is well known that a longer duration of an open wound and delayed referral increase the risk of infection, the impairment of ischemia, the size of the wound, and the depth of the wound [7]. In the current study, wound duration of greater than 4 weeks at the time of assessment was an independent factor associated with hospitalization. The data support the hypothesis that severely complicated DFUs may result from delayed referrals of uncomplicated or complicated DFUs that are not adequately diagnosed or managed, leading to unavoidable impairment and tissue loss. This key point reinforces the need for early diagnosis, early treatment, and potential application of the fast-track model, which aims to clearly identify the characteristics of any DFUs and the specific timing of referrals according to the grade of wound severity. Early diagnosis and an adequate pathway result in a reduction in the cases of late referral and in the chances of a negative wound evolution in patients with a newly active DFU [7,21,22,23].

The aims of the fast-track model are to identify DFU severity and to prioritize the treatment of each kind of DFU, especially the treatment of severely complicated DFUs. These DFUs should be evaluated by a dedicated DFS within 24 h. In addition, the fast-track model aims for the early identification and adequate management of less critical cases, such as uncomplicated and complicated DFUs, to avoid the risk of progression to severely complicated DFUs. The fast-track model provides a simple clinical evaluation tool that includes a patient’s clinical history, general health status, foot pulses, and signs of infection, as well as a probe-to-bone procedure, which can be easily used without specialist tools. The fast-track model can be utilized by all professionals working within primary care, such as GPs, diabetologists, nurses, and podiatrists, to achieve the identification of wound severity. The association found between the grade of DFU severity identified by the fast-track model and the outcomes of interest provides HCPs in primary care with a common language and a prioritization tool to support their decision making. In summary, we believe the fast-track model may support the following:

(a) easy communication among expert HCPs working in different DFSs, including community DFSs and hospitals;

(b) easy evaluation of DFU severity by clinical evaluations (including evaluations of patients’ general health status and DFUs), without requiring specialized settings;

(c) assessment of the prognosis in terms of the DFUs’ severity, as reported in this study;

(d) defining the specific timing of referrals to specialized DFSs, according to a DFU’s severity;

(e) targeting treatment according to each specific clinical scenario, based on IWGDF guidelines.

The current study is a monocentric study undertaken in a tertiary level DFS. Therefore, local policy differences and training may lead to bias in other centers where another similar pathway may be adopted, based on possible population differences and different patient presentations. Even so, the fast-track model has been developed to promote integrated management between primary care providers and dedicated DFSs. In light of differences in healthcare professional training, healthcare organizations, and legislation in different countries, the main principles of the fast-track model can be adapted to local needs and local healthcare organizations.

Accordingly, this study may be greatly enhanced by a multicentric study, in which DFSs operating in primary care are included.

Nonetheless, in the present study, the management of patients has been considered in the context of applicable diabetic foot guidelines. Some variables that may influence outcomes, such as PAD severity in ischemic patients, effectiveness of revascularization procedures, and wound location, have not been evaluated.

## 5. Conclusions

There is a close association between DFU severity, as identified by the fast-track model, and outcomes of interest, such as healing, amputation, hospitalization, and survival. By stratifying the prognosis of a DFU, primary health care providers using this simple tool will know the potential prognosis of any DFU, as documented by the fast-track model, and, consequently, avoid any delay in referring patients to specialized DFSs.

## Figures and Tables

**Table 1 jcm-12-00761-t001:** Definition of three levels of DFU severity, according to the fast-track model. DFU: diabetic foot ulcer; ESRD: end-stage renal disease.

Uncomplicated DFU	Complicated DFU	Severely Complicated DFU
▪ Superficial ▪ Not ischemic ▪ Not infected	▪ Ischemic or suspected ischemia (absence of pulses, presence of necrosis or dry gangrene) ▪ Infected (mild/moderate) ▪ Deep (tendons, muscles and/or bone) ▪ Any kind of ulcers in patients with ESRD on dialysis and/or heart failure	▪ Wet gangrene ▪ Necrotizing fasciitis ▪ Abscess ▪ Presence of fever ▪ Clinical signs of sepsis

**Table 2 jcm-12-00761-t002:** Baseline demographic, clinical, and wound characteristics of the whole population and the sub-groups defined by DFU severity. Data are n (%) or means ± SEM or %. * *p* < 0.05 vs. uncomplicated DFU; † *p* < 0.05 vs. complicated DFU. DFU: diabetic foot ulcer; HbA1c: glycated hemoglobin; IHD: ischemic heart disease; ESRD: end-stage renal disease.

Variable	Whole Population (n = 367)	Uncomplicated DFU (n = 35)	Complicated DFU (n = 210)	Severely Complicated DFU (n = 122)	*p* Value (X^2^-ANOVA)
Age (years)	69 ± 13	67 ± 12	69 ± 13	70 ± 16	0.3
Sex (male)	230 (62.7%)	21 (60%)	128 (61%)	81 (66.4%)	0.06
Diabetes (type 2)	344 (93.7%)	30 (85.7%)	198 (94.3%)	116 (95.1%)	0.2
Diabetes duration (years)	19 ± 10	11 ± 5	18 ± 9 *	23 ± 12 * †	0.002
HbA1c	7.7 ± 1	7.3 ± 0.7	7.5 ± 0.6	8.2 ± 1.8 * †	0.004
Hypertension	353 (95.9%)	31 (88.6%)	202 (96.2%)	120 (97.5%)	0.1
Dyslipidemia	291 (79.3%)	21 (60%)	167 (79.5%)	103 (84.4%)	0.01
IHD	169 (46%)	4 (11.4%)	101 (48.1%) *	64 (52.4%) * †	<0.0001
Heart failure	82 (22.3%)	0 (0%)	51 (24.3%) *	31 (25.4%) *	0.003
ESRD	45 (12.3%)	0 (0%)	21 (10%) *	24 (19.7%) * †	0.009
Smoking	42 (11.4%)	5 (14.3%)	21 (10%)	16 (13.1%)	0.4
Foot ischemia	210 (57.2%)	0 (0%)	130 (61.9%) *	80 (65.6%) *	<0.0001
Infection	286 (77.9%)	0 (0%)	164 (78.1%) *	122 (100%) * †	<0.0001
Deep ulcer (to the bone)	248 (67.6%)	0 (0%)	131 (62.4%) *	117 (95.9%) * †	<0.0001
Ulcer Size (>5 cm^2^)	242 (65.9%)	11 (31.4%)	117 (55.7%) *	114 (93.4%) * †	<0.0001
Gangrene	170 (46.3%)	0 (0%)	91 (43.3%) *	78 (63.9%) * †	<0.0001
Wound duration (weeks)	5 ± 1.5	2 ± 0.5	4 ± 1	8 ± 3 * †	0.03

**Table 3 jcm-12-00761-t003:** Outcomes of interest for the whole population and for patients with uncomplicated, complicated, and severely complicated diabetic foot ulcers. DFU: diabetic foot ulcer.

Variable	Whole Population (n = 367)	Uncomplicated DFU (n = 35)	Complicated DFU (n = 210)	Severely Complicated DFU (n = 122)	Severely Complicated DFU vs. Uncomplicated DFU		Severely Complicated vs. Complicated DFU	
					HR (95% CI)	*p* Value	HR (95% CI)	*p* Value
Healing	300 (81.7%)	34 (97.1%)	181 (86.2%)	85 (69.8%)	0.4 (0.3–0.9)	0.0001	0.6 (0.4–0.9)	0.002
Minor Amputation	124 (33.8%)	1 (2.9%)	42 (20%)	81 (66.4%)	12.5 (4.5–19.6)	<0.0001	5.1 (2.3–9.1)	<0.0001
Major Amputation	26 (7.1%)	0 (0%)	6 (2.9%)	20 (16.4%)	8.3 (2.6–11.5)	<0.0001	3.3 (1.8–6.5)	<0.0001
Hospitalization	231 (62.9%)	5 (14.3%)	117 (55.7%)	109 (89.3%)	15.6 (3.5–30.8)	<0.0001	6.1 (2.8–10.1)	<0.0001
Survival	347 (94.5%)	35 (100%)	203 (96.7%)	109 (89.3%)	0.5 (0.2–0.8)	<0.0001	1.2 (0.8–1.8)	0.08

**Table 4 jcm-12-00761-t004:** Multivariate analysis of independent predictors of outcomes (healing, major amputation) found via univariate analysis. Peripheral arterial disease, DFU severity, and osteomyelitis were independent predictors of healing; end-stage renal disease and DFU severity were independent predictors of major amputation. DFU: diabetic foot ulcer.

Variables	Healing			Major Amputation		
	OR	95% CI	*p* Value	OR	95% CI	*p* Value
End-stage renal disease	1.2	0.5–1.3	0.1	2.3	1.8–4.6	0.0003
Ulcer size (>5 cm^2^)	1.4	0.6–1.8	0.08			
Infection	0.9	0.8–1.5	0.06	1.3	0.9–1.5	0.08
Peripheral arterial disease	0.5	0.2–0.8	0.001	1.6	1.2–2.5	0.09
DFU severity (severely complicated DFU vs. complicated DFU)	0.3	0.1–0.7	<0.0001	3.1	1.5–5.9	<0.0001
Osteomyelitis	0.4	0.2–0.9	0.04			
Wound duration (>4 weeks)	1.5	0.9–1.6	0.3			

**Table 5 jcm-12-00761-t005:** Multivariate analysis of independent predictors of outcomes (survival and hospitalization) found via univariate analysis. End-stage renal disease and DFU severity were independent predictors of survival; heart failure, infection, and peripheral arterial disease, DFU severity, sepsis, and wound duration were independent predictors of hospitalization. DFU: diabetic foot ulcer.

Variables	Survival			Hospitalisation		
	OR	95% CI	*p* Value	OR	95% CI	*p* Value
End-stage renal disease	0.5	0.3–0.8	0.001			
Heart failure				1.8	1.2–2.2	0.03
Infection				2.5	1.1–5.6	0.002
Peripheral arterial disease				3.5	1.8–7.5	0.0001
DFU severity (severely complicated DFU vs. complicated DFU)	0.2	0.1–0.5	0.001	12.5	3.8–18.7	<0.0001
Sepsis				8.3	2.1–9.2	<0.0001
Wound duration (>4 weeks)				1.4	1.1–4.2	0.01

## Data Availability

Data are unavailable due to privacy and ethical restrictions.

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
