# Peer review of "Validation of the Fast-Track Model: A Simple Tool to Assess the Severity of Diabetic Foot Ulcers"

_jcm, 2023, doi:10.3390/jcm12030761_

Round 1

Reviewer 1 Report

I thank the authors for exploring an important area of diabetic foot research. This research aimed to evaluate and validate the association between the grade of diabetic foot ulcer severity proposed by the fast-track model and outcomes of interests such as healing, amputation, hospitalization, and survival. The current study found evidence of association(s) between the grade of DFU severity proposed by the fast-track model and considered outcomes. Overall, this manuscript conveys the main message clearly. However, for publication, the following issues need to be considered.

Abstract

1.     I am not sure if the paper's title closely aligns with this study's aim. Authors may reconsider the title.

2.     Line 23: It would be better to mention the specific outcome clearly.

3.     Line 23-24: mention sample size and patients’ geographical location. International readers need to know this information.

4.     Line 30: authors can remove “Three-hundred sixty-seven patients were included,” as it will be addressed in comment 3.

5.     Line 31-35: The result part of the abstract needs to be rewritten. For example, I am not clear about the numbers 97.1%, 86.2%, 69.8% (p<0.0001) for healing. Is it “97.1%, 86.2%, and 69.8% patients had uncomplicated, complicated and severely complicated DFU, respectively? If so, it needs to be clearly written. What does the p-value signify?

6.     A conclusion or policy implication is missing in the abstract.

Materials and Methods

7.     Information on patients’ geographical location and tertiary-level DFS is needed.

8.     Line 69: What did the authors mean by “reduced life expectancy”? What was the cut-off point for exclusion?

9.     A description of outcomes of interest needs to be provided in the Method section.

10.  What is the basis of selected variables in the univariate and multivariate analyses? Authors need to justify the inclusion of variables in the statistical model based on previous literature.

11.  I would be interested to know more about the ethical review statement of this study. An explanation of the waiver needs to be provided.

12.  Line 136-137: “personal computer”, confusing?

13.  Line 140: the word is “univariate”. Please consider using it in the whole manuscript.

14.  Line 144-145: The aim of this study is to assess the relationship between diabetic foot ulcer severity and outcomes of interest, not to predict outcomes. Therefore, I think that mentioning this is not necessary.

15.  Please consider citing appropriate references for the statistical analysis.

Results

16.  While section 3.1 provides the necessary information from Table 2, it is overwhelmed with too many statistical figures. I would recommend rewriting section 3.1. For example, authors may consider the use of “most of”, “more than three-quarters”, etc types of adjectives to describe the results.

17.  The title of Table 2 is, in fact, a table footnote. I would recommend authors provide a proper title.

18.  It is not clear to me what the authors meant by X anova in the last column of Table 2.

19.  Third last row: size of what (ulcer, I guess)?

20.  *P<0.05 vs. Uncomplicated DFU: based on which statistical test? Needs to be mentioned. Similarly, †P <0.05 vs. Complicated DFU: based on which statistical test?

21.  Similar to the earlier comment, a better title for Table 3 is required. HRs can be presented in two separate columns.

22.  Please consider presenting results in the sequence of OR, 95% CI, and p-values (although p-values are not required if 95% Cis are presented) in Table 4 and Table 5.

23.  Results in Tables 4 and 5 need to be elaborated in terms of ORs.

24.  Reference groups for each variable need to be mentioned in Table 4 and Table 5, either in the main table or as a footnote.

25.  Although not important, no statistics related to Shapiro-Wilk test were shown in the result section. Authors may consider removing this statistical analysis from the method section or explain briefly by presenting relevant figures.

26.  Overall, I would suggest authors provide the full form of all abbreviations in table footnotes as well.

Reviewer 2 Report

This retrospective study evaluates the outcome of diabetic foot ulcers (DFU) which are divided in three severity groups.

It is shown, that more complicated DFU are followed by higher amputation rates and higher mortality.

The statistic analysis seems reasonable.

However there are some drawbacks:

Materials and Methods:

Reduced life-expactancy were excluded. (line 69+70)

One could argue that patients with Necrotizing fasciitis or sepsis like in severly complicated DFU have also a reduced life-expactancy.

This should be clarified.

Co-Morbidities (line 79):

Smoking should be evaluated as a co-morbidity and should be included in the multivariante regression analysis.

Table 1:

Severly complicated DFU are wound infections of a DFU.

However the severity of a Necrotizing fasciitis (NF) or a sepsis is remarkably higher compared to an abscess. Therefore urgency and treatment differ between these cases.

In line 127: … within 24 hours.

As mentioned above sepsis and NF are life threatening situations which require immediate treatment, and cannot prolonged for 24 hours.

Results:

Table 2:

Sepsis is not a variable here, because if signs of sepsis occur it is defined as severly complicated DFU.

Therefore sepsis should be excluded in this table.

Table 3:

In severly complicated DFU a healing rate of 69.8% is given.

In my opinion healing means, that the ulcer is closed (by conservative or operative treament).

But in this group 66.4% needed minor and 16.4% major amputations.

Does that mean, that healing is also when a amputation is well healed? This hast o be clarified.

Table 4 and 5 are not referred to in the manuscript.

In Table 4 and 5 different ORs are lacking. Is this for a specific reason?

Discussion:

In the discussion limitations of the study have to be discussed.

In line 190: severe complicated DFU are a predictor for survival. Is this a mistake? Is it non-suvival respectively mortality?

Line 206+207: Higher infection rates are found in severly complicated DFU.

Severly complicated DFU are defined according to Table 1 as infected DFU or DFU with systemic infection signs. Therefore they will always have 100% infection rates.

Summary (line 244 and follwoing):

This study describes the outcome of different severities of DFU. However the study nor the classification itself cannot give advise  for the targeted treatment of DFU. Because in the classification different aspects of DFU like osteomyelitis, vascular impairment etc. are not taken into account.

Line 277 – 281:

Many co-authors are mentioned regarding a retrospective analysis of 378 patients.

However there are drawbacks of this study.

The group of uncomplicated DFU is small (n=35).

This might be due  to the fact, that it is a monocentric study of a tertiary centre. This must be discussed. The study would strongly benefit by a multicentric design, for example with a primary hospital to include more uncomplicated DFU.

In Introduction is writen: 

The FTP has been acceppted by scientific community as a practical model...

However this statement is verified by two self-citations. Additional citations would be more convincing.

Author Response

This retrospective study evaluates the outcome of diabetic foot ulcers (DFU) which are divided in three severity groups.

It is shown, that more complicated DFU are followed by higher amputation rates and higher mortality.

The statistic analysis seems reasonable.

However there are some drawbacks:

Materials and Methods:

Reduced life-expactancy were excluded. (line 69+70)

One could argue that patients with Necrotizing fasciitis or sepsis like in severly complicated DFU have also a reduced life-expactancy.

This should be clarified.

Regarding this point, authors agree with the reviewer concerning that sepsis and necrotizing fasciitis increase the risk of mortality. Anyway, reduced life-expectancy was referred in the specific case only to patients with high risk of mortality in the next 6 months due to the general health status, regardless the new foot problem, and/or often suitable only for a conservative therapy. This point has been now better described for being clearer.

Co-Morbidities (line 79):

Smoking should be evaluated as a co-morbidity and should be included in the multivariante regression analysis.

Smoking has been now included among the clinical variables (see methods section, co-morbidities). Anyway, it did not appear to be an independent predictor of outcome at the univariate analysis.

Table 1:

Severly complicated DFU are wound infections of a DFU.

However the severity of a Necrotizing fasciitis (NF) or a sepsis is remarkably higher compared to an abscess. Therefore urgency and treatment differ between these cases.

Dear Reviewer, agree on your comment. Necrotizing fasciitis can require urgent surgical treatment and the pattern may be more aggressive than an abscess. Nonetheless, the idea of the fast-track is to involve in a specific setting of severity the diabetic foot infection which require urgent treatment such as necrotizing fasciitis or sepsis and those which require urgent evaluation (and/or treatment) to avoid local and general impairment such as abscess, phlegmons and wet gangrene. In the current manuscript authors have reported a previous classification on DFU severity developed by the fast-track pathway.

In line 127: … within 24 hours.

As mentioned above sepsis and NF are life threatening situations which require immediate treatment, and cannot prolonged for 24 hours.

According to this right comment, the sentence has been revised reinforcing the concept that necrotizing fasciitis and sepsis should be immediately referred.

Results:

Table 2:

Sepsis is not a variable here, because if signs of sepsis occur it is defined as severly complicated DFU.

Therefore sepsis should be excluded in this table.

As suggested, the variable “sepsis” has been removed.

Table 3:

In severly complicated DFU a healing rate of 69.8% is given.

In my opinion healing means, that the ulcer is closed (by conservative or operative treament).

But in this group 66.4% needed minor and 16.4% major amputations.

Does that mean, that healing is also when a amputation is well healed? This hast o be clarified.

Dear Reviewer, thank you for this remark. This point has better clarify in methos section as follows: “Healing was defined as complete wound healing without major amputation, including both patients who required a minor amputation for achieving wound healing and those who recovered without a minor amputation”

Table 4 and 5 are not referred to in the manuscript.

Thank you for this note. They have been added in the text

In Table 4 and 5 different ORs are lacking. Is this for a specific reason?

To be concise, in table 4 and 5 all independent predictors found at univariate analysis for all the outcomes of interest are reported in the same table. Where the OR is lacking in the row, it is due to the fact that this specific variable did not result an independent predictor for the specific outcome reported in the column. i.e in table 5, osteomyelitis was an independent predictor of healing at the univariate analysis, but not significant for major amputation. For this reason, the variable “osteomyelitis” is not reported in the column of “major amputation” at the multivariate analysis.

Discussion:

In the discussion limitations of the study have to be discussed.

Study limitations have been reported inside the discussion section (see lines 270-279). In addition, it has been added the following sentence: “the study may greatly benefit from a multicentric study in which also DFS operating in primary care should be included”

In line 190: severe complicated DFU are a predictor for survival. Is this a mistake? Is it non-suvival respectively mortality?

The authors mean that severely complicated DFU negatively influence the chance of survival, anyway “survival “has been modified in “mortality” to be clearer. Thank you for the advice.

Line 206+207: Higher infection rates are found in severly complicated DFU.

Severly complicated DFU are defined according to Table 1 as infected DFU or DFU with systemic infection signs. Therefore they will always have 100% infection rates.

Agree with this remark. Anyway, author think to report this data to reinforce as infection was more present in severely complicated DFU in comparison to complicated DFU, although also complicated DFU included patients with mild/moderate infection. Hope this reply can be clear for the reviewer.

Summary (line 244 and follwoing):

This study describes the outcome of different severities of DFU. However the study nor the classification itself cannot give advise  for the targeted treatment of DFU. Because in the classification different aspects of DFU like osteomyelitis, vascular impairment etc. are not taken into account.

The aim of the fast-track model is to consider all the item which can influence the severity of DFUs, including suspected osteomyelitis and vascular impairment which are included directly in the “severely complicated DFU”. See table 1 and lines 251-254 (“The fast-track model provides a simple clinical evaluation tool which includes clinical history, general health status, foot pulses, signs of infection, probe-to-bone procedure, etc. that can be easily examined without specialist tools”).

Regarding the targeted treatment, the fast-track model recommends any kind of treatment in the respect of International Guidelines (International Working Group on the diabetic foot). See also lines 268-269: “…targeted treatment according to every specific clinical scenario based on IWGDF guidelines”.

Line 277 – 281:

Many co-authors are mentioned regarding a retrospective analysis of 378 patients.

Concerning this point, the authors included in the study are all the members of International diabetic foot care group and even if the majority are not involved in the collecting data, all of them discussed on the aim, approved the design and developed the project.

However there are drawbacks of this study.

 The group of uncomplicated DFU is small (n=35).

This might be due  to the fact, that it is a monocentric study of a tertiary centre. This must be discussed. The study would strongly benefit by a multicentric design, for example with a primary hospital to include more uncomplicated DFU.

Dear reviewer, agree on this comment and proposal. This point has been included in the limitations section inside the discussion. Agree also regarding the group with uncomplicated DFU resulted small in comparison to complicated and severely complicated DFU, anyway this data reflects more or less literature data where ischaemic and infected ulcer reported a higher prevalence than neuropathic and non-infected DFU, being present both in more than 60% of cases.

 In Introduction is writen: 

The FTP has been acceppted by scientific community as a practical model...

However this statement is verified by two self-citations. Additional citations would be more convincing.

Dear reviewer, thank you for this comment which allows to better clarify this point. The self-citation was referred to the effectiveness of the fast-track pathway recently documented by a retrospective study, while the authors mean that the document has been also accepted as practical tool by the D-Foot international among scientific projects. To reinforce this point, a reference inside the text has been added (a link to connect directly to D-Foot international homepage). See lines 58-60: “The FTP has been accepted by scientific community as practical model to identify a specific pathway according DFU’ severity (see https://d-foot.org/projects/fast-track-pathway-for-diabetic-foot-ulceration)”.

Round 2

Reviewer 2 Report

The suggested corrections were made.

However, the limitations of the study persists. They are now more cleearly discussed.